# Assessment of Red Blood Cell Aggregation in Preeclampsia by Microfluidic Image Flow Analysis—Impact of Oxidative Stress on Disease Severity

**DOI:** 10.3390/ijms25073732

**Published:** 2024-03-27

**Authors:** Anika Alexandrova-Watanabe, Emilia Abadjieva, Ina Giosheva, Ariana Langari, Tihomir Tiankov, Emil Gartchev, Regina Komsa-Penkova, Svetla Todinova

**Affiliations:** 1Institute of Mechanics, Bulgarian Academy of Sciences, 1113 Sofia, Bulgaria; anikaalexandrova@abv.bg (A.A.-W.); abadjieva@gmail.com (E.A.); tiho_bg@abv.bg (T.T.); 2University Obstetrics and Gynecology Hospital “Maichin Dom”, 1431 Sofia, Bulgaria; ina_gi@abv.bg (I.G.); egartt@gmail.com (E.G.); 3Institute of Biophysics and Biomedical Engineering, Bulgarian Academy of Sciences, 1113 Sofia, Bulgaria; arianalangari@abv.bg; 4Department of Biochemistry, Medical University-Pleven, 5800 Pleven, Bulgaria; regina.komsa-penkova@mu-pleven.bg

**Keywords:** preeclampsia, red blood cell aggregation, microrheological properties of the blood, microfluidic device, image flow analysis, oxidative stress

## Abstract

Preeclampsia (PE) is a hypertensive disease characterized by proteinuria, endothelial dysfunction, and placental hypoxia. Reduced placental blood flow causes changes in red blood cell (RBC) rheological characteristics. Herein, we used microfluidics techniques and new image flow analysis to evaluate RBC aggregation in preeclamptic and normotensive pregnant women. The results demonstrate that RBC aggregation depends on the disease severity and was higher in patients with preterm birth and low birth weight. The RBC aggregation indices (EAI) at low shear rates were higher for non-severe (0.107 ± 0.01) and severe PE (0.149 ± 0.05) versus controls (0.085 ± 0.01; *p* < 0.05). The significantly more undispersed RBC aggregates were found at high shear rates for non-severe (18.1 ± 5.5) and severe PE (25.7 ± 5.8) versus controls (14.4 ± 4.1; *p* < 0.05). The model experiment with in-vitro-induced oxidative stress in RBCs demonstrated that the elevated aggregation in PE RBCs can be partially due to the effect of oxidation. The results revealed that RBCs from PE patients become significantly more adhesive, forming large, branched aggregates at a low shear rate. Significantly more undispersed RBC aggregates at high shear rates indicate the formation of stable RBC clusters, drastically more pronounced in patients with severe PE. Our findings demonstrate that altered RBC aggregation contributes to preeclampsia severity.

## 1. Introduction

Preeclampsia, a hypertensive multisystem disorder, affects 3–5% of pregnancies and stands as a prominent contributor to maternal and neonatal morbidity and mortality, with its occurrence primarily in the later stages of pregnancy. The clinical diagnostic criteria involve elevated blood pressure (systolic blood pressure ≥ 140 mm Hg and diastolic blood pressure ≥ 90 mm Hg) after the 20th week of gestation in women whose blood pressure had previously been within the normal range. In most cases, PE is accompanied by proteinuria (defined as protein excretion ≥ 0.3 g in a 24-h urine specimen) as well as renal insufficiency, liver dysfunction, pulmonary edema, visual or cerebral disturbances, and/or thrombocytopenia [1]. PE can also lead to circulation problems with insufficient blood flow to the placenta and thus limit the fetus’s supply of nutrients and oxygen and consequently growth restriction and low birth weight (LBW).

The pathophysiology of preeclampsia remains unclear, but it is considered that an abnormal formation of the placenta may serve as the triggering factor [2]. The trophoblast invasion in preeclampsia is impaired, reducing placental perfusion and creating intermittent flow [3]. Various stress factors can disturb the syncytiotrophoblast; however, the main one responsible for preeclampsia is uteroplacental malperfusion secondary to defective remodeling of the uterine spiral arteries. The failed remodeling of extravillous trophoblasts to deeply invade the uterine spiral arteries and to replace the vascular endothelial cells leads to hypoxia and reduced uteroplacental blood flow to the placenta [4]. Chronic placental hypoxia leads to electron slip from oxidative phosphorylation, reduced ATP production, and the overproduction of reactive oxygen species (ROS), thus inducing damage to both the placenta and the endothelium [5]. Oxidative stress of the syncytiotrophoblasts is one of the characteristic features of preeclampsia especially of its early onset. In response to syncytiotrophoblast stress, soluble factors (such as antiangiogenic factors, cell-free nucleic acids, and pro-inflammatory cytokines) and microvesicles are released from the placenta into the maternal bloodstream [6,7,8]. The imbalance of proangiogenic and antiangiogenic factors in maternal circulation plays a key role in triggering PE. It has been confirmed that soluble fms-like tyrosine kinase 1 (sFLT1) and soluble endoglin (sEng) are significantly increased in preeclamptic patients compared to normotensive pregnant women. At the same time, concentrations of vascular endothelial growth factor (VEGF) and placental growth factor (PlGF) are decreased [9]. Evidence suggests that the interaction of soluble factors with maternal endothelium contributes to vascular dysfunction [10]. The disruption of endothelial function results in a systemic inflammatory response, which is one of the clinical syndromes of pre-eclampsia [11].

Red blood cells (RBCs) are highly susceptible to oxidative stress and inflammation, making them among the first cells to be affected by unfavorable conditions. Throughout their life span, they are exposed to internal and external sources of oxidants and inflammatory cytokines that can impair their primary function of delivering oxygen to tissues. RBCs, as carriers of respiratory gases, are continuously exposed to a high oxygen concentration, which promotes ROS production. RBCs are also exposed to oxidants derived from the surrounding endothelial cells and cells of the immune system. To protect against the effects of ROS, RBCs are equipped with antioxidant machinery. Under normal conditions, enzyme systems and low molecular weight antioxidants largely prevent red blood cell damage [12]. However, in some pathological situations and hereditary diseases that create conditions for excessive oxidative stress, the redox balance is disturbed, and the RBC protective mechanisms become ineffective [13,14]. Weakening or deficiencies in the RBCs’ antioxidant systems can lead to cellular damage. Several studies have reported elevated RBCs and plasma glutathione peroxidase in severe preeclampsia [15,16]. As a result of excess levels of oxidants, red blood cells can undergo modifications in both lipid and protein components, including lipid peroxidation and concomitant externalization of phosphatidylserine (PS). Since membrane lipids of RBCs are rich in polyunsaturated fatty acids, their exposure to ROS damages membrane stability and causes hemolysis. The process of lipid peroxidation can disturb the integrity of the membrane, thus leading to deviations in its fluidity and permeability [17]. Lipid peroxidation products alter RBC membrane functions by cross-linking the skeletal proteins of the RBC membrane and lead to protein polymerization, potassium leakage from the cell, and RBC dehydration. Oxidative modifications of cytoskeletal proteins, Band 3, and cytoskeleton-membrane interactions can largely affect red blood cell deformability and membrane stability [18,19,20]. Spickett et al. have demonstrated increased RBC lysis in preeclamptic patients compared to normotensive pregnant women as a result of oxidative damage to the RBC membrane, leading to reduced fluidity [21].

RBC properties have a strong influence on blood flow. RBC fluidity and elasticity enable them to deform as they pass through narrower vessels. Thus, they withstand hemodynamic changes in blood flow, ultimately reducing vascular resistance. The ability of RBCs to adapt their morphology in response to shear forces is the main determinant of blood viscosity. In normal conditions, in large vessels where blood velocity is low, with a lower shear rate (<1–10 s^−1^), red blood cells tend to aggregate (forming so-called rouleaux or 3-dimensional clusters), promoting a rise in apparent whole-blood viscosity [22]. The conditions of the formation of aggregates largely determine the rheological behavior of the blood, especially in the microcirculation. RBC aggregation is reversible and shear dependent. At high shear rates (>150 s^−1^) in large vessels, RBCs’ deformability and dissociation result in low whole-blood viscosity [23]. High shear forces (i.e., in the arterial system) induce the disaggregation and dispersion of RBCs thereby facilitating microcirculation [24]. RBC aggregation depends on various factors, including local shear rate, hematocrit, plasma concentration of high molecular weight proteins as well as RBC properties such as surface charge, membrane fluidity, and cell deformability. Increased concentration of fibrinogen and other high molecular weight proteins in plasma was found to be related to increased red blood cell aggregation [25]. Exposure of RBCs to oxidative stress also affects both the extent and strength of their aggregation. ROS may damage RBCs by increasing the strength of their aggregation, thus leading to an elevation in vascular resistance against blood flow, especially at the precapillary level [26]. A change in RBC aggregation can be an important indicator of disease as well as of the course of disease progression. Therefore, the understanding of the extent of aggregation and its effect on blood rheology is important as it can be enhanced in some pathological states and diseases including infections [27], diabetes mellitus [28], cardiovascular disease [29], hypertension [30], preeclampsia [31], and other pathologies.

Microcirculation can be seriously disturbed in case of deviation of elasticity and deformability of RBCs. RBC aggregates with impaired deformability are known to be difficult to disperse as the shear rate increases [32]. Abnormalities in the shape or deformability of red blood cells leading to stiffening of the cell membrane in some inherited diseases such as sickle cell anemia, hemolytic anemias, and thalassemic syndromes affect whole-blood viscosity [33]. Previous studies have found that RBC deformability is reduced in women with PE and in some cases of intrauterine growth restriction and contributes to reduced microcirculation [34]. The increased viscosity due to enlarged RBC aggregation found in patients with PE may impair proper tissue perfusion in the placental intervillous space. Damaged RBCs enhance the progression of endothelial dysfunction and maternal circulatory disorders [35,36].

In recent years, the developments of microfluidics technologies with microchannels that simulate dimensions of the smallest human capillaries have allowed the study of rheological and mechanical properties of RBCs and their influence on blood flow in health and disease [37,38]. Microfluidics technology is often used to mimic microcirculation and microvasculature networks. In microfluidics, data is usually collected optically using a high-speed video system connected to a microscope and a micropump [39]. The alteration of RBC aggregation at different shear rates in a microfluidic system provides valuable information for the variations in blood rheology. Recent advances in microfluidic systems have led to more realistic models of blood vessels, used in many studies for blood flow analysis and appropriate selection of experimental flow conditions. These models applied the shear force effects on cell dynamics and their use in microfluidic applications [40,41].

To evaluate the main indices of RBC aggregates, several image processing and analysis techniques have been applied. In their study, Foresto et al. quantified the RBC aggregation at stasis and under several shear stresses to better mimic aggregation behavior in vivo by defining an aggregation shape parameter [42]. Fusman et al. (2000) have found that the area between adjacent RBC aggregates, known as the vacuum radius, is a direct indicator of RBC aggregate size [43]. Using this concept, Kaliviotis et al. (2011) and Dusting et al. (2009) have defined an indirect index of aggregation based on RBC-free area, hematocrit, and total area have been determined from microscopic blood images [44,45]. A computerized image analysis technique has been applied to determine the size distribution of RBC aggregates at different shear stresses by varying the flow rate of suspensions in flow chambers [46]. Kavita and Ramakrishnan (2007) use a two-dimensional wavelet transform to find the aggregation size index as the standard deviation of the coefficients generated by the wavelet functions [47].

In this study, a microfluidic-based measurement was applied to investigate RBC aggregation at low and high shear rates in normal pregnancies and PE patients and evaluate the relationship between the severity of preeclampsia and the extent of RBC aggregation. A new image processing methodology was applied to evaluate RBC aggregation indices (AI). We revealed that the RBC aggregation is significantly elevated in patients’ samples vs. the control ones, especially in cases of severe PE. The in vitro-induced oxidative stress in red cells from healthy donors demonstrates that the elevated AI in PE RBCs can be partially due to the effect of the oxidants.

## 2. Results

### 2.1. Characteristics of the Patients with Preeclampsia and Healthy Pregnant Women

In this study, thirteen pregnant women suffering from preeclampsia were enrolled. Of the surveyed cases, six had severe preeclampsia requiring emergency delivery, and the rest had a mild form of PE (referred to as non-severe PE). The non-severe cases of PE were defined in the presence of hypertension (blood pressure > 140/90) but below 160 systolic BP and 110 diastolic BP, with or without the presence of proteinuria and in the absence of significant end-organ dysfunction. The severe PE cases were defined as PE with the presence of systolic blood pressure of 160 mmHg or higher or diastolic blood pressure of 110 mmHg and/or proteinuria of more than 3+ on 2 random urine samples collected at least 4 h apart and/or significant end-organ dysfunction (persistent headaches, epigastric pain, elevated liver enzymes, thrombocytopenia, etc.). The main clinical characteristics and laboratory indices from PE patients and normotensive pregnant women are summarized in Table 1, Appendix A for individual patients. The mean maternal age at the time of blood sampling of severe PE was significantly above that of the PC and non-severe PE. The systolic and diastolic blood pressures were statistically higher in the PE groups than in the control pregnant group. Premature births occur significantly more in 69% of studied PE cases. The weight of babies born to mothers with non-severe PE and severe PE was 16% and 59%, respectively, less than that of newborns of mothers with normal pregnancies.

The value of CRP was within a wide range both for PE patients and controls. However, it should be noted that CRP exceeded the upper limit in 42% of non-severe PE and 66% of severe PE cases, and only in 25% of the controls. Serum creatinine and ASAT tended to have higher values in patients with preeclampsia, but these values did not reach statistical significance. All other laboratory indices did not differ between the groups (Table 1).

According to the gestational age at the time of diagnosis, 54% of the investigated PE cases were of early onset. All cases with severe PE were early onset, while for those with non-severe PE, only one case was early onset.

### 2.2. RBC Aggregation and Main Rheological Indices of Healthy Pregnant Women and Patients with Preeclampsia

Red blood cells from the blood of patients with PE were isolated at the time of diagnosis before patients received any treatment. RBC aggregation was monitored at two conditions—at low-shear conditions, followed by a high-shear flow for the PE and control groups. The obtained images are shown in Figure 1. At a low shear rate (8.9 s^−1^) in the RBC suspension from normotensive women, small, stacked aggregates called “rouleaux” formed by a small number of RBCs were observed (Figure 1A, Figure 2A and Appendix A). Data examination shows that the “rouleaux” formations are the predominant part of the RBC aggregates (Appendix A). The image analysis of the PE RBCs revealed a difference both in the size and the number of aggregates depending on disease severity (Figure 1B,C, Appendix A).

RBCs from the non-severe PE group formed predominantly larger branched aggregates (Figure 2B and Appendix A) compared to the control group (Figure 1A,B). The image analysis determined that the RBC aggregation index at a low shear rate (AI_L_) was significantly higher (by 26%, *p* = 0.04) compared to that of the PC group (Table 2). The average number of RBC aggregates at a low shear rate (NA_L_) in the non-severe PE subgroup was lower (by 23%) than in PC; however, we did not find a statistically significant difference (Table 2). Therefore, the non-severe PE subgroup formed larger, but fewer, RBC aggregates than the control group of healthy pregnant women.

In the severe PE subgroup, larger aggregates are formed, having extended branches (Figure 2C and Appendix A), with which they are connected in aggregate complexes, forming networks, in comparison with non-severe PE cases and PC group (Figure 1). RBC aggregation in severe PE cases (Figure 1C) was increased compared to both the non-severe PE subgroup and normotensive pregnant women (Figure 1A,B). The RBC aggregation index (AI_L_) increased statistically significantly in severe PE cases by 75% (*p* = 0.04) compared to the control group, and by 39% compared to the non-severe PE subgroup at a low shear rate. The number of RBC aggregates (NA_L_) was significantly higher (by 38%) compared to the non-severe form of PE (*p* = 0.005, Table 2) and slightly above that of the PC group. Therefore, in the severe PE subgroup, RBCs were connected into large complex aggregates, more numerous compared to both the control group and non-severe PE cases.

As mentioned above, in healthy status the RBC aggregation process is reversible under high shear flow conditions. Thus, we analyzed RBC behavior under a high-shear rate (446 s^−1^), simulating the microcirculation in the narrow capillaries/arterioles. From image analysis, it was estimated how many of the formed RBC aggregates did not break up under high shear flow conditions (Figure 3). In the RBC suspension of the control group, a few undispersed aggregates are observed, formed by a small number of cells (Figure 3A). In cases of non-severe PE and especially in severe PE, a greater number of small RBC aggregates consisting of more RBCs remain after high-flow conditions than in healthy pregnant women (Figure 3B,C).

In the PC group, NA_H_ under high-flow conditions was reduced by more than 88% and AI_H_ by 94% relative to a low shear rate (Table 2). Undispersed RBC aggregates from non-severe PE were statistically significantly more (by 26%, *p* = 0.049) compared to the PC group (Table 2). However, the AI_H_ value for the non-severe PE subgroup showed no statistical difference from that of the controls. In contrast, both aggregation indices of the severe PE subgroup were significantly higher, more than two times (*p* = 0.007 for AI_H_ and *p* = 0.005 for NA_H_) compared to the corresponding PC values (Table 2). Therefore, as the preeclampsia worsens, the degree of RBC aggregation increases.

### 2.3. Rheological Features of RBCs under In Vitro Chemically Induced Oxidative Stress

It is known that oxidative stress is involved in the pathogenesis of preeclampsia and has significant impacts on RBCs. To assess the degree of influence of oxidative stress on the rheological changes of red blood cells and its relationship with pathological conditions during pregnancy, we examined the effect of hydrogen peroxide (H_2_O_2_) on newly isolated RBCs from three healthy pregnant women, not included in the main PC group. H_2_O_2_ is a metabolite involved in most of the redox metabolism processes of living cells and is, therefore, often used in model systems [48]. The number of RBC aggregates and the RBC aggregation index are determined after treatment of the cells with three concentrations of H_2_O_2_ (Table 3).

At a low shear rate, treatment of red blood cells with a low concentration of H_2_O_2_ (200 mM) did not induce a shift to increased aggregation compared to non-treated cells. The aggregates observed were similar to those formed at low shear rates, such as untreated RBCs (Figure 4A,B). No difference was found between the values of the respective NA_L_ indices of the untreated samples and those treated with 200 mM H_2_O_2_. At the low concentration of H_2_O_2_, AI_L_ was statistically significantly reduced (by 15%, *p* < 0.05) compared to that of non-treated RBCs (Table 3).

As the concentration of H_2_O_2_ (300 mM) increases, RBC aggregation improves. Larger aggregates with visible branching were observed (Figure 4C). The AI_L_ aggregation index was significantly increased (by 63%, *p* < 0.05) above that of untreated RBCs (Table 3) and was comparable to that of severe PE (Table 2).

At the highest concentration of H_2_O_2_ (400 mM), the RBC aggregation further increased (*p* < 0.05) and even exceeded that of the severe PE group. The AI_L_ of RBCs treated with 400 mM H_2_O_2_ is twice as large as untreated cells, while NA_L_ was nearly 1.6 times lower (Table 3) due to the greatly increased aggregates’ size (Figure 4D). The image analysis (Figure 4C,D) of the RBCs treated with the two high concentrations of H_2_O_2_ (300 mM and 400 mM H_2_O_2_) showed a strong similarity of the aggregates formed (Figure 4C,D) with those corresponding to non-severe and severe PE (Figure 1B and Figure 1C, respectively).

At a high shear rate, the disaggregation of the formed aggregates in the untreated RBC suspension was visible (Figure 5A). Compared with the RBC aggregation index of untreated RBCs, AI_H_ was increased (by 25%) when cells were treated with 200 mM H_2_O_2_ (Table 3). Aggregate dispersion in suspensions from treated cells was weak compared to untreated; several small aggregates could be seen especially in 300 mM and 400 mM H_2_O_2_ solution (Figure 5C,D). However, the aggregation indices, AI_H_ and NA_H_, were not different from those of the untreated group.

At the lowest concentration of H_2_O_2_ tested (200 mM), the aggregation index of red blood cells was lower, while the number of aggregates was not significantly different from that of untreated cells. The increasing H_2_O_2_ concentration at a low shear rate leads to an elevation in both AI and the number of RBC aggregates compared to those of the untreated ones (Table 3). The Pearson’s correlation analysis we applied demonstrated a strong relation (r = 0.9; *p* = 0.033) for AI_L_ of the pair parameters of treated cells with 300 mM H_2_O_2_ and those derived from the non-severe PE group and moderate (r = 0.55; *p* = 0.05) for RBCs treated with 400 mM H_2_O_2_ and RBCs from the severe PE group. Although the oxidative stress model we applied in this study does not fully account for disease progression, it sheds light on the influence of oxidative stress on the RBC rheological behavior.

At a high shear rate, the aggregation index of RBCs treated with the three H_2_O_2_ concentrations did not differ significantly from that of untreated cells. At treatment with 200 and 300 mM H_2_O_2_, the number of undispersed RBC aggregates at high shear flow conditions was lower than that of RBCs treated with the highest concentration of H_2_O_2_ (400 mM), and of untreated ones. We hypothesize that this effect is a reflection of intact antioxidant defense machinery in healthy individuals.

## 3. Discussion

Assessing of rheological changes in the RBC features plays a key role in understand-ing complex pathophysiological processes affecting the microcirculation in preeclampsia. The main manifestation of PE- hypertension is associated with abnormalities in blood viscosity [49], which strongly depends on the aggregation and deformability of red blood cells [50].

Based on the concept that preeclampsia is a multisystem disorder, it can be classified into different subtypes. According to the time of its manifestation, PE is defined as early-onset type and late-onset preeclampsia. These two main types of preeclampsia (early-onset and late-onset) are reported to have different risk factors and pathophysiology but lead to a common manifestation. However, all women with pre-eclampsia can be at risk of deterioration regardless of the timing of disease onset. Thus, considering the severity of clinical symptoms and the involvement of organ dysfunction, PE can be stratified into severe or no-severe PE [51].

### 3.1. Changes in the Rheological Features of RBCs in Preeclampsia

In our investigation, we classified PE patient samples according to their clinical characteristics as severe and non-severe PE and then compared their corresponding rheological features. Determining the disease severity was based on a complex assessment of the patient’s status, which includes, on the one hand, the presence of increased blood pressure and/or proteinuria, but also dysfunction in some organs, vertigo, headaches, edema, etc. The RBC aggregation, a mechanism that greatly influences the non-Newtonian properties of blood, is the main determinant of low shear blood viscosity [52]. For all studied patients, the values of AI_L_ of RBC aggregates at low shear rates were found to be statistically increased relative to those of the PC group, although to a different extent. To quantitively relate the disease severity with the increased RBC aggregability, a correlation approach was applied. Pearson’s correlation analysis between patients’ blood pressure and the corresponding RBC aggregation indices supports our finding that RBC aggregation correlates with disease severity (Appendix A). A very strong correlation was found between the respective values of AI_L_ and the diastolic BP of non-severe PE patients (r = 0.87) and the systolic BP of patients with severe PE (r = 0.81), and a strong correlation between the AI_L_ and systolic BP of non-severe PE (r = 0.61) and diastolic BP of severe PE group (r = 0.65), respectively.

Another criterion we applied to differentiate the rheological properties of patients’ RBCs was their rheological behavior at high shear-flow conditions. Taken together, the changes in these pairs of parameters, combined with the image analysis used to determine the number and size of the aggregates, allowed us to analyze the cases of the two PE subsets for their association with disease severity and the time of PE onset (early and late PE).

For more than half of the PE cases, forming the first PE subset, i.e., non-severe PE, the main rheological indices at low shear rate, AI_L_, and the number of RBC aggregates differ from the controls showing enlarged RBC aggregation. Image analysis of the non-severe PE cases revealed the formation of long-chain rouleaux, unlike controls where these rouleaux were shorter and less branched. At a high shear rate, the number of the RBCs′ aggregates was higher than the control ones possibly as a consequence of enhanced cell adhesion forces. However, the values of AI_H_ were similar to those of the PC group, indicating the reversibility of the aggregates. This finding suggests that the disease has a moderate, but still distinguishable effect on RBCs relative to the PC group. Indeed, this group included non-severe cases of PE, most of them being late-onset preeclampsia and one case of early-onset. In their work, Pepple et al., who investigated only mild preeclampsia, determined no altered RBC aggregation in preeclamptic compared to non-preeclamptic women [35]. This discrepancy with our results is probably due to different techniques used and, on the other hand, to the methodology for determining aggregation (aggregation index vs. the aggregation half time).

A more significant deviation in RBC rheological characteristics from the control values was found for the remaining PE cases. This subset included samples from patients with severe preeclampsia. It is worth noting that all these patients were of early-onset PE. Regarding RBC aggregation indices, the severe PE group strongly differs from the PC group both at low and high shear rates, indicating the formation of stable RBC clusters that are difficult to break down at high flow conditions. Consistent with our results, Heilmann et al. found statistically elevated red cell aggregation at stasis and low shear rates, along with increased levels of hematocrit and hemoglobin in patients with severe PE compared to normal pregnant women [31]. However, a certain influence of the time of onset of PE should not be excluded since all patients with severe forms of preeclampsia are also of an early onset. Recently, Csiszar et al. reported an increased RBC aggregation in women with early-onset PE but did not specify the severity of symptoms [36]. The enhanced formation of immense RBC aggregates we observed in the severe PE group, even in the absence of plasma protein enhancing RBC aggregation, suggested their robust impact on the patients′ microcirculation. Red blood cell aggregation largely determines the viscosity of blood at low-shear conditions, such as the placental intervillous space. Altered RBC aggregation would impair placental microcirculation, and, in these conditions, the RBCs would not be able to accomplish their main function of oxygen delivery [53]. The elevated RBC aggregation could influence tissue hypoxia in PE. In this regard, rouleaux was found to be present in pathological conditions involving hypoxia [54].

Among the factors influencing elevated RBC rouleau formations is the bridging action of plasma proteins, especially fibrinogen. In the cases we tested, the levels of fibrinogen did not differ from the controls; on the other hand, the experiments were carried out with washed RBCs which suggested that the observed rheological alterations can be attributed to the changes in the RBC biophysical features, which probably play a significant role in PE. For example, membrane fluidity of RBCs, an important factor in modifying cell rheology, was found to be reduced in subjects with essential hypertension [55]. Changes in the lipid composition of RBC plasma membranes in hypertensive individuals have been found to result in morphological and physiological abnormalities that alter the dynamic properties of RBCs [56].

Blood viscoelastic properties are particularly sensitive indicators of aggregation and stiffness of RBCs. At a low shear rate (<10 s^−1^), the viscoelastic properties depend mainly on the RBC aggregation [57]. Our results demonstrated enhanced RBC aggregation in PE cases, especially in the severe group relative to the controls. Therefore, the elevated RBC aggregation in PE groups could be caused by the impaired viscoelasticity of RBCs in PE patients.

Increased RBC aggregation may result from conformational alterations of membrane components occurring during hypertension. These changes in turn lead to membrane stiffness and loss of elasticity. The extensive decrease of RBC membrane elasticity or deformability in some pathological conditions is indicative of abnormal cell morphology and damaged RBC capacity to adequately adapt its shape to navigate blood vessels, resulting in impaired hemorheology [58]. Recently, we have reported impaired membrane integrity and the presence of RBCs with unusual morphology in preeclamptic women [59]. A great percentage of schistocytes (fragmented red blood cells) was also found in peripheral blood smears of the samples from patients with hypertensive disorders of pregnancy [60]. It is notable that under hypoxic conditions, partial oxygenation of Hb takes place, increasing autoxidation, which leads to ROS production near the RBC membrane. Aiming to better understand the impact of hypoxia and oxidative stress on PE pathology, we studied the impact of in vitro chemically induced oxidative stress on RBCs.

### 3.2. Effect of a Chemically Induced Oxidative Stress

Oxidative stress plays an important role in many pathological conditions and has been recognized as one of the factors in the pathogenesis of PE. Oxidative stress is accepted as one of the determinants of RBC aggregation, which increases with the increased level of reactive oxygen species (ROS) [26,61,62]. To examine the degree of impact of ROS on the rheological RBC alterations, we mimicked in vitro oxidative stress with H_2_O_2_ on newly isolated RBCs from normotensive pregnant donors. Hydrogen peroxide is generated by most cells in the human body and is present in blood at low concentrations. An interesting finding, we made was that at low concentrations of H_2_O_2_ (200 mM), the degree of aggregation and aggregation indices were somewhat less expressed compared to untreated cells. In line with this finding, it was previously demonstrated a slightly increased RBC membrane deformability in response to H_2_O_2_ treatment [63]. As noted above, RBC deformability largely determines the formation of aggregates, altering rheological blood properties. Another plausible hypothesis is that low-dose oxidant incubation triggers the innate defense RBC mechanisms that are intact in healthy cells and elicit a potent response.

The proposed model showed that the pretreatment with an increasing concentration of H_2_O_2_ (300 and 400 mM) led to changes in the degree of aggregation and the main rheological indices of RBCs, similar to those observed in the cells of both PE groups. The strong correlation we found in the aggregation index between 300 mM and 400 mM H_2_O_2_ treated cells and that of women with PE, supports the notion that ROS contributes to the hyperaggregation of red blood cells. Recently, our oxidative stress model on RBCs showed a significant decrease in biconcave cells (young cells) at the expense of senescent cells, i.e., spiculocytes and spherocytes [64]. It is reported that the aggregation of old cells is much higher than that of young cells due to the reduced electrostatic repulsion and markedly improved cell-cell affinities of senescent RBCs [65].

The RBCs possess multiple enzymatic and non-enzymatic antioxidant defense mechanisms to prevent oxidative damage. However, during persistent oxidative stress, these mechanisms may become exhausted [66,67]. According to research by N. A. Besedina et al. (2022), the built-in antioxidant defense system has a limit exceeding which hemoglobin oxidation, membrane, and cytoskeleton transformation occurs. It leads to cell swelling, increased stiffness, and adhesion, which in turn results in a decrease in the transit velocity in microcapillaries [68]. At low levels of oxidative stress, the biophysical properties of “healthy” red blood cells do not change significantly, while at high levels, only a small part of RBCs remains stable with intact biophysical properties and retain their ability to move in microcapillaries [68]. In addition, overexpressed oxidative stress can cause lipid peroxidation, changes in their morphology, disturbance in membrane-cytoskeleton linkage, or conformation alteration of membrane proteins [69,70]. Such alterations of RBC characteristics affect the tendency of RBC to aggregate and consequently to impaired rheological properties. For example, excessive oxidative stress induces membrane reorganization leading to the exposure of PS to the outer surface. PS externalization, in turn, is associated with increased RBC aggregation [71]. In their study, Balaji et al. have demonstrated that externalization of phosphatidyl serine induces the appearance of hydrophobic regions on the plasma membrane, increasing its stickiness and thus enhancing red blood cell aggregability [72]. A strong correlation was found between the amounts of malonyldialdehyde (MDA), the product of lipid peroxidation, and the increased viscosity values of the suspensions of oxidized RBCs at low shear rates, due to a higher tendency of RBCs to form aggregates under oxidative stress [73]. Increased membrane lipid peroxidation is characteristic of several RBC diseases, such as sickle cell disease, thalassemia, unstable hemoglobin disease, etc. In addition, placental ischemia in preeclampsia has been found to reduce the antioxidant activity of glutathione peroxidase and catalase, enzymes, responsible for the degradation of H_2_O_2_ in RBCs [74]. Levels of antioxidants in the blood of women with preeclampsia have been demonstrated to be reduced, leading to oxidative modifications of RBC membrane proteins [75].

Upon exposure to large amounts of oxidant, hemoglobin, as the primary target of oxidative damage, can undergo spontaneous oxidation of iron in its heme groups to form methemoglobin. On the other hand, oxidized hemichromes preferentially bind to the cytoplasmic domain of band 3, thereby inducing the formation of band 3 aggregates. The clustering of Band 3 protein significantly reduces the main RBC functions, including their antioxidant capacity [76].

Our results also demonstrated that the dispersion of RBC aggregates at high-flow conditions was not significantly affected by the H_2_O_2_ treatment. This finding supports the assumption that the RBC antioxidant systems of healthy donors are not damaged and can resist external influences maintaining normal circulation in the microcapillaries, in contrast to pathological ones.

## 4. Materials and Methods

### 4.1. Study Groups and Ethics Statement

The study population consisted of 8 pregnant controls (designated as PC) in the third trimester of pregnancy (mean age 26.2 ± 5.6 years) and 13 pregnant women with PE (seven patients with non-severe PE, mean age 27.4 ± 5.8 years, and six patients with severe PE, 36.8 ± 4.3 years) admitted to the Medical University, Pleven or the Hospital of Obstetrics and Gynecology “Maichin Dom”, Medical University Sofia. In addition, three other normotensive pregnant women (mean age 26.7 ± 4.0 years) were recruited to isolate RBCs for the subsequent oxidative stress study. The diagnosis and the severity of PE were determined according to hypertension in pregnancy guidelines [77]. Women in the control group had no complications during pregnancy or elevated blood pressure, and all delivered after 38 weeks of gestation.

Exclusion criteria: Pregnant women with chronic hypertension, thyroid diseases, diabetes, kidney diseases, erythrocytopathies, autoimmune diseases, hyperlipidemia, or fetal malformations were not included in this study.

All women involved in the investigation provided written informed consent The study was approved by the Ethics Committee of Medical University-Pleven (approval No. 404-KENID 22 October 2015) and was performed following the Helsinki International ethical standards on human experimentation.

### 4.2. Sample Preparations

Blood samples were drawn by venipuncture in 6 mL tubes (Vacutainer; Becton Dickinson, and Company, Franklin Lakes, NJ, USA) containing K_2_EDTA. According to Baskurt et al., EDTA is the most widely used anticoagulant in hemorheological studies [78]. Blood from patients with preeclampsia was collected immediately after diagnosis, before treatment was prescribed, and by normotensive pregnant women at their prenatal visits. The RBCs were isolated by centrifugation from freshly drawn blood (centrifuge Universal 320 R, Hettich, Germany) at 1500 g for 10 min, and the yellowish supernatant (plasma and white blood cells) was discarded. The remaining RBCs were resuspended and washed three times in PBS solution (140 mM NaCl, 2.7 mM KCl, 8 mM Na_2_HPO_4_, 1 mM KH_2_PO_4_), pH 7.4. The hematocrit of the washed RBC suspension was adjusted to 40% (centrifuge Haematokrit 200, Hettich, Tuttlingen, Germany) in PBS buffer.

### 4.3. Preparation of Oxidized RBCs

Oxidative stress was chemically induced using hydrogen peroxide (H1009 Sigma-Aldrich Pty Ltd., an affiliate of Merck KGaA, Darmstadt, Germany) on newly isolated RBCs from three healthy pregnant women who were not included in the control group PC according to the protocol described in [59]. In brief, RBCs were treated for 4 h at 37 °C with 200 mM, 300 mM, and 400 mM H_2_O_2_. The reaction was stopped with 200 μL 10 mM EDTA. The oxidized RBCs were resuspended and washed two times in PBS solution (140 mM NaCl, 2.7 mM KCl, 8 mM Na_2_HPO_4_, 1 mM KH_2_PO_4_), pH 7.4. Oxidized RBC suspensions were diluted to 40% hematocrit in PBS buffer for further investigation.

### 4.4. Stimulation RBC Aggregation

For the stimulation of RBC aggregation, 10 μL from RBC suspensions (hematocrit 40%) from PE and PC groups’ samples, and those treated with hydrogen peroxide is taken and added to 200 μL of Dextran 70 solution (at a concentration of 4 g/dL) to stimulate RBC aggregation. The final hematocrit of the thus diluted RBC suspensions decreased to 2% for all experiments.

### 4.5. Viscosity Measurements

The viscosity of the diluted RBC suspensions in Dextran 70, prepared according to the methodology described in Section 4.4. was determined in steady state flow conditions by a Brookfield Programmable Viscometer DV-II+Pro (Brookfield Engineering Laboratories, Inc., Middleboro, MA, USA). The device is calibrated with water at a temperature of 37 °C. The measured viscosity of the diluted suspensions of RBCs in Dextran 70 at a temperature of 37 °C is 1.12 ± 0.04 mPa.s.

### 4.6. Microfluidic System and Experiments

The RBC aggregation was studied using the air pressure-driven microfluidic system BioFlux (Fluxtion Biosciences, Oakland, CA, USA), a high-quality imaging platform for conducting in-flow rheological analyses. The microfluidic system consists of a BioFlux 200 electro-pneumatic flow control pump; BioFlux microfluidic plates; inverted fluorescence microscope Lumascope 620; and computer configuration with operational and analysis software. For experiments assessing RBC aggregation, BioFlux plates 24-well 0–20 dyn/cm^2^, containing 8 microfluidic channels with cross-sectional dimensions of 350 μm in width and 75 μm in height were used.

### 4.7. Design of the Experiments

The microfluidic channels were filled with 200 μL of each prepared diluted RBC suspension in Dextran 70. Two modes of operation are used. In the first one, the RBC suspension was initially perfused through the microfluidic channels at a high shear stress of 5 dyn/cm^2^ (corresponding to a shear rate of 446 s^−1^), for 5 min to disperse pre-existing aggregates, following which, the flow in the channels was abruptly reduced to a low shear stress of 0.1 dyn/cm^2^ (corresponding to a shear rate of 8.9 s^−1^) for 5 min.

After stopping the flow, the formed RBC aggregates were imaged using an inverted microscope operating in the Phase Contrast mode.

In the second mode, the aggregates formed in the channel were subjected to shear stress of 5 dyn/cm^2^ (shear rate 446 s^−1^), again for 5 min. The images of the remaining RBC aggregates not destroyed by the flow were imaged using the Phase Contrast mode of the microscope.

After low and high shear rates (each of them with a duration of 5 min), 30 s after stopping the flow, at least 5 images were taken every 2 s along the entire visible length of the channel at randomly selected locations.

To evaluate the aggregation of RBCs in each sample, the mean values (±SD) of the rheological parameters detected from all images taken were used.

### 4.8. Computational Image Analysis for the Evaluation of the RBC Aggregates

To evaluate RBC aggregates in the current study, a new methodology was developed, which includes an experimental method and a computer algorithm based on it. The elaborated algorithm, shown as a flowchart illustrating the processing steps, and the corresponding resulting images are shown in Figure 6.

The algorithm for the assessment and analysis of the obtained image data is illustrated in Figure 6A. The obtained raw images (Figure 6B) of RBC aggregates are imported as input data into the Image J 1.54 g software environment. The selection of the criteria parameter is based on the elaborated experimental methodology and the application of the image intensity transformation process and the region segmentation analysis. Contrast Software Setting (image intensity transformation process) (Figure 6C) is an automatic process, that includes such variation of the pixel values of the obtained images to achieve improvement of visibility and to prepare the image for further analysis. The second step of this process is the realization of the region segmentation analysis, whereby the application of the incorporated in the Image J tool—threshold, the image data results in dividing the image into regions based on the intensity values of its pixels (Figure 6C). Hence, the processed after the contrast software automatic setting image is converted into a binary image, where pixels are classified into two categories: foreground (object of interest—RBC aggregates) and background.

The next step is the selection of the criteria parameter, which determines the separation between RBC aggregates and non-aggregates. The differentiations of the state of the cells (aggregate or non-aggregate) were determined according to the number of the stuck together/(clumped) cells one over the other. We have chosen as an initial parameter (minimum) 3 grouped cells, according to Yeom E. and Lee S.J. [79]. Based on this parameter, the initial assessment of RBC aggregates is performed. To refine this algorithm, a final evaluation parameter is set. It verifies that all the RBC aggregates are selected (Figure 6D). If this parameter is fulfilled/met, the next step is to obtain the table with results for the size of the area and the number of each object of interest. In case this is not accomplished, the process of RBC aggregates evaluations is repeated. A detailed description of the quantification of the number of RBC aggregates (NA) is given in Appendix A.

The obtained results for the number and the area size of the RBC aggregations were further analyzed in a program elaborated in IntelliJ IDEA.

### 4.9. RBC Aggregation Measurement

After image analysis, RBC aggregation indices are determined. Two indices of RBC aggregation were measured: (i) the extent of RBC aggregation (RBC aggregation index, AI) and (ii) the number of RBC aggregates (NA), at low (8.9 s^−1^) and high shear rates (446 s^−1^).

The first *AI_L_* represents the power of RBC aggregation under low-flow conditions, calculated by the formula adapted by [80]:$${AI}_{L}=\frac{S_{2}}{S_{1}}$$
where *S*_2_ (in pixels) is the total sum of areas of the aggregates (after the flow of 8.9 s^−1^), and *S*_1_ (in pixels) is the total observed area of one visual field of the microscope. Each image (with area of *S*_1_) in Figure 1, Figure 3, Figure 4, Figure 5 and Figure 6 represent 1600 × 300 pixel^2^.

The second *AI_H_* presents the not-destroyed RBC aggregates at high-flow conditions and is calculated by the formula:$${AI}_{H}=\frac{S_{3}}{S_{1}}$$
where *S*_1_ (in pixels) is the total observed area of one visual field of the microscope (1600 × 300 pixel^2^), and *S*_3_ (in pixels) is the total sum of areas of RBC aggregates at high shear rate (after the flow of 446 s^−1^).

### 4.10. Statistics

Data were presented as mean ± SD (standard deviation). A non-parametric Wilcoxon test was used to compare data between groups. Significant differences were registered at the level of *p* ≤ 0.05. Pearson’s correlation analysis was performed to determine the correlation between patients’ blood pressure and corresponding RBC aggregation indices, and between AI_L_ of H_2_O_2_ treated cells and untreated RBCs.

## 5. Conclusions

The obtained results revealed significantly altered rheological behavior both in low and high shear flow conditions of red blood cells obtained from women with preeclampsia as compared to that of healthy gestational age-matched pregnant women. This change is reflected in (i) RBCs from PE patients becoming visibly significantly more adhesive, forming large, branched aggregates at low shear rate; (ii) increased RBC aggregation index in the patients’ cells; (iii) significantly more undispersed RBC aggregates at a high shear rate indicating the formation of stable RBC clusters, which are difficult to break down at high flow conditions. In addition, the modifications found in RBCs are drastically more pronounced in patients with severe PE, thus let us assume their contribution to disease severity. Therefore, the assessment of the rheological properties of red blood cells could serve as an additional criterion for distinguishing the disease severity.

The applied in vitro model of oxidative stress suggests the involvement of ROS in the pathological changes in PE RBCs and/or their reduced antioxidant protective mechanisms.

Our findings could provide new insights into the complex mechanisms of preeclampsia and help shape future preventive strategies.

## Figures and Tables

**Figure 1 ijms-25-03732-f001:**
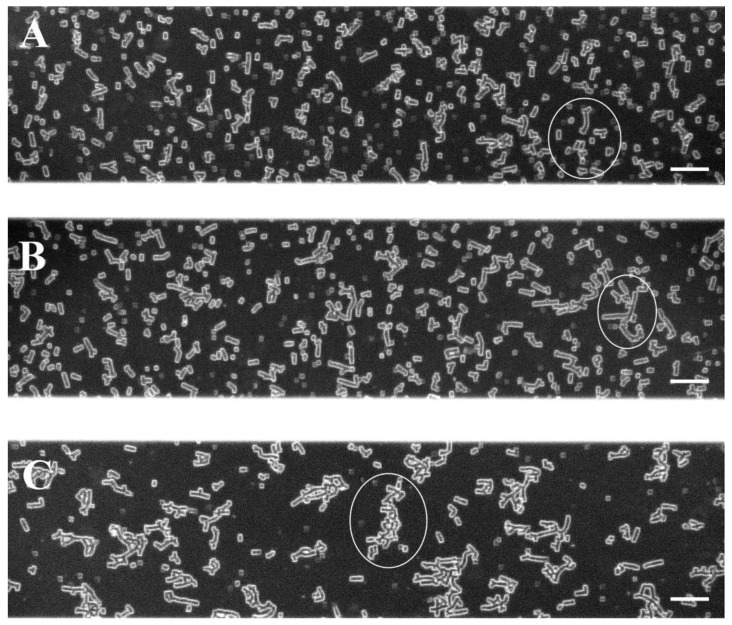
Representative images of RBC aggregates obtained with the BioFlux microfluidic system under low-flow condition (8.9 s^−1^), from (**A**) healthy pregnant women (n = 8); (**B**) patients with non-severe preeclampsia (n = 7), and (**C**) patients with severe preeclampsia (n = 6). Scale bar—50 µm. The encircled RBC aggregates (examples are depicted with white circles in **A**–**C**) are shown enlarged in Figure 2.

**Figure 2 ijms-25-03732-f002:**
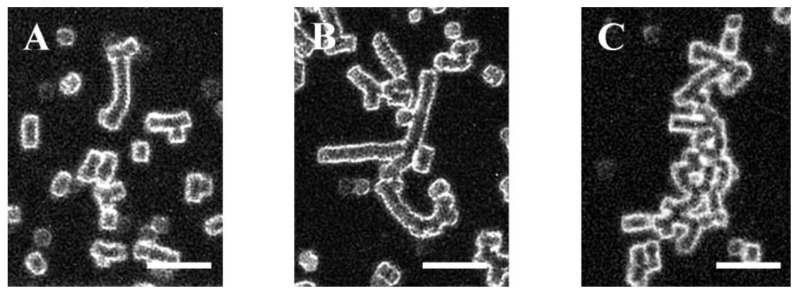
Representative images of RBC aggregate shapes: the “rouleaux” formation (**A**) as identified in Figure 1A, and different types of aggregates designated as branched aggregates (**B**) and extended branches, i.e., network (**C**) as identified in Figure 1B,C, respectively. Scale bar—25 µm.

**Figure 3 ijms-25-03732-f003:**
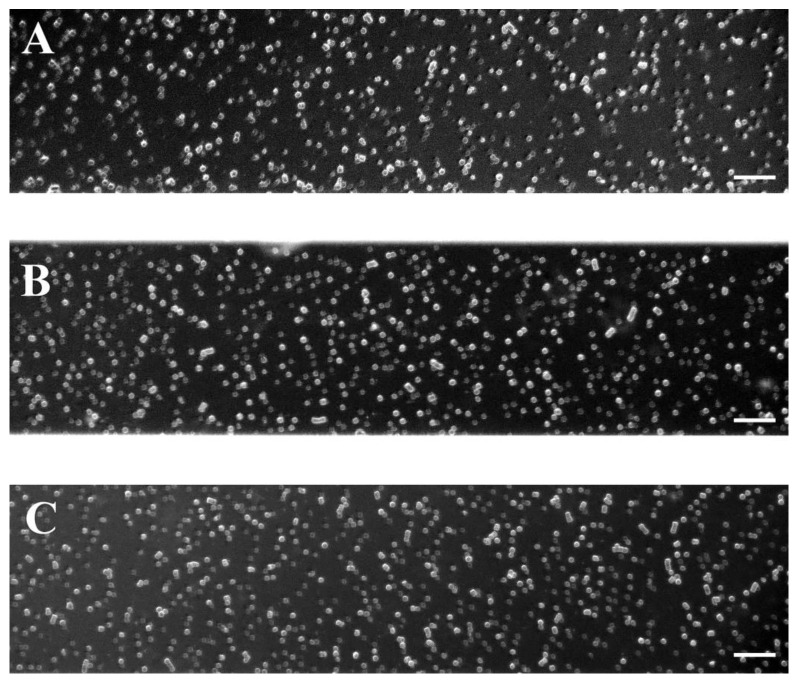
Representative images of RBC aggregates obtained with the BioFlux microfluidic system under high-flow conditions (after the flow of 446 s^−1^), from: (**A**) healthy pregnant women (n = 8); (**B**) patients with non-severe preeclampsia (n = 7), and (**C**) patient with severe preeclampsia (n = 6). Scale bar—50 µm.

**Figure 4 ijms-25-03732-f004:**
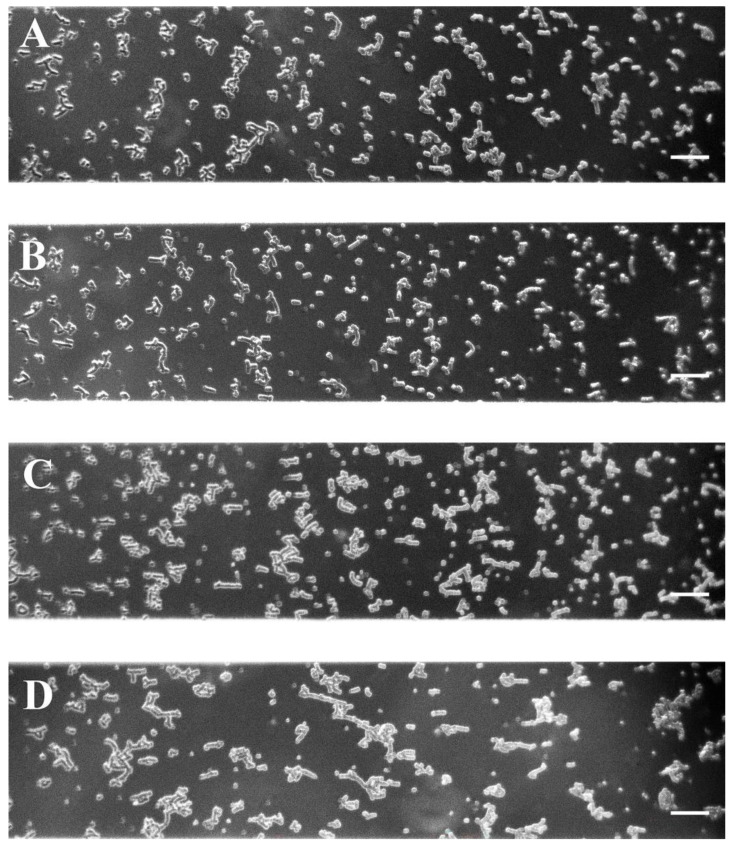
Representative images of RBC aggregates obtained with the BioFlux microfluidic system under low-flow conditions (after the flow of 8.9 s^−1^) for untreated cells derived from healthy pregnant women (n = 3) (**A**) and cells subjected to 200 mM (**B**), 300 mM (**C**) and 400 mM (**D**) H_2_O_2_ treatment. Scale bar—50 µm.

**Figure 5 ijms-25-03732-f005:**
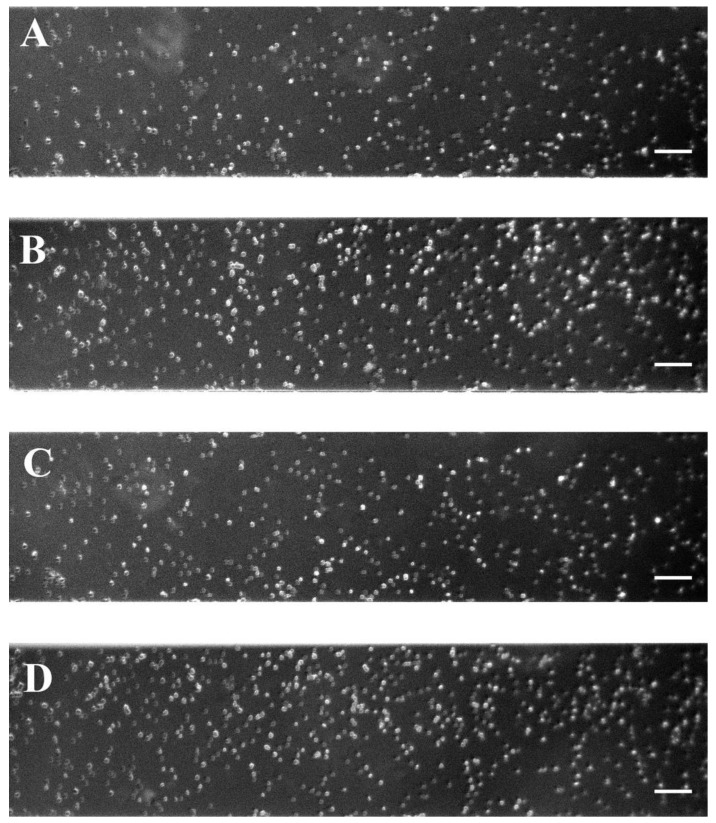
Representative images of RBC aggregates obtained with the BioFlux microfluidic system under high-flow conditions (after the flow of 446 s^−1^) for untreated cells derived from healthy pregnant women (n = 3) (**A**) and cells subjected to 200 mM (**B**), 300 mM (**C**), and 400 mM (**D**) H_2_O_2_ treatment. Scale bar—50 µm.

**Figure 6 ijms-25-03732-f006:**
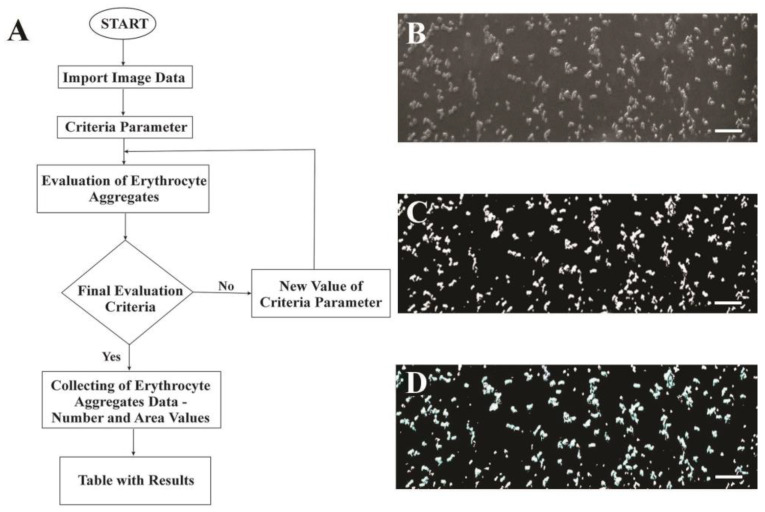
Analysis of the bioimage data: (**A**) Block-scheme of an algorithm for RBC aggregate analysis; (**B**) raw image of RBC aggregates taken with BioFlux system, using a 10× magnitude objective and Phase Contrast mode; (**C**) image after applying the segmentation process and fulfilling the final evaluation criteria; (**D**) image of selected RBC aggregates (circled in blue), which was used to determine their number and area. Scale bar—50 µm.

**Table 1 ijms-25-03732-t001:** Clinical data (maternal age; gestational age, GA at the time of diagnosis and at delivery; mean blood pressure, BP; newborn weight, low body weight, LBW) and laboratory indices (Proteinuria, PU; RBC count; Hemoglobin, Hb; Hematocrit, Ht; mean corpuscular volume in femtoliters (fl), MCV; mean corpuscular hemoglobin, MCH; mean corpuscular hemoglobin concentration, MCHC; red cell distribution width, RDW; Fibrinogen, Fg; C-reactive protein (CRP); Platelet Count; Creatinine serum; aspartate aminotransferase, ASAT; and alanine aminotransferase, (ALAT) of the pregnant controls (PC), and women with preeclampsia (PE).

Characteristics	Reference Values for Pregnant Women	PC (n = 8)	Non-Severe PE (n = 7)	Severe PE (n= 6)
Maternal age (years)		26.2 ± 5.6	27.4 ± 5.8	36.8 ± 4.3 *
Mean BP (systolic/diastolic)		105 ± 7/70 ± 5	145 ± 3/94 ± 6 *	165.5 ± 4/112 ± 2 *
GA at diagnostic of PE			34.8 ± 3.4 *	28.8± 1.3 *
GA at delivery		39.1 ± 1.1	35.3 ± 3.0 *	29.3 ± 1.0 *
Newborn weight (g)		3366.7 ± 124.7	2824.3 ± 416.0 *	1388.0 ± 356.6 *
Number of LBW		-	1	6
Proteinuria (mg in 24-h urine collection)	-	-	644 ± 6 *	1975 ± 646 *
Number of patients with PU			3	5
Fg (g/L)	2.90–6.50	5.7 ± 0.2	5.6 ± 0.4	5.7 ± 1.1
CRP (mg/L) (interval)	0.5–5.0	4.2–19.9	2.00–26.87	2.37–76.82
Creatinine serum (µmol/L)	35–80	66.25 ±9.1	74.4 ±7.3	76.0 ±15.1
ASAT (U/L)	4–32	16.3 ± 0.9	29.2 ± 17.2	33.5 ± 20.8
ALAT (U/L)	3–30	16.6 ± 4.3	26.1 ± 18.3	19.7 ± 8.5
RBC count (T/L)	3.6–5.1	3.84 ± 0.18	3.94 ± 0.3	3.87 ± 0.4
Hb (g/L)	110–148	118.7 ± 10.7	122.2 ± 6.8	118.2 ± 5.8
Ht (L/L)	0.30–0.46	0.37 ± 0.03	0.36 ± 0.02	0.36 ± 0.02
MCV (fl)	82–98	94.05 ± 3.13	92.50 ± 3.73	92.60 ± 6.47
MCH (Pg/L)	26.5–32.0	31.0 ± 1.8	31.0 ± 1.4	30.7 ± 2.4
MCHC (g/L)	295–360	329.5 ± 9.23	345.7 ± 24.2	330.8 ± 6.5
RDW %	12.3–14.7	12.9 ± 1.3	13.6 ± 1.6	13.6 ± 2.1
Platelet Count ×10^9^/L	146–429	249.5 ± 63.2	269.2 ± 68.2	220.8 ± 99.1

* Indicates statistically significant difference (*p* < 0.05) in the PE values for both severe and non-severe groups compared with the PC values.

**Table 2 ijms-25-03732-t002:** RBC Aggregation Indexes (AI) and Number of RBC Aggregates (NA) at low (AI_L_ and NA_L_) and high shear rates (AI_H_ and NA_H_) of the patients with non-severe (n = 7) and severe preeclampsia (n = 6) (non-severe PE and severe PE) and the control group of healthy pregnant women (n-8) (PC group). Mean values and SD.

Groups	Low Shear Rate		High Shear Rate	
	AI_L_	NA_L_	AI_H_	NA_H_
PC	0.085 ± 0.01	119.4 ± 38.7	0.005 ± 0.002	14.4 ± 4.1
Non-severe PE	0.107 ± 0.01 *	92.1 ± 16.7	0.006 ± 0.001	18.1 ± 5.5 *
Severe PE	0.149 ± 0.05 *	127.0 ± 9.1 ^♦^	0.011 ± 0.002 *^♦^	25.7 ± 5.8 *

* Indicates statistically significant difference (*p* < 0.05) in the PE values for both severe and non-severe groups compared with the PC values; ^♦^ Indicates statistically significant difference (*p* < 0.05) in the values of the severe PE group compared with the respective values of the non-severe PE group.

**Table 3 ijms-25-03732-t003:** RBC aggregation indexes (AI) and number of RBC aggregates (NA) at low (AI_L_ and NA_L_) and high shear rates (AI_H_ and NA_H_) of the non-oxidized, newly isolated RBCs from healthy pregnant donors and cells treated with three concentrations of hydrogen peroxide (H_2_O_2_). Mean values and SD.

Groups	Low Shear Rate		High Shear Rate	
	AI_L_	NA_L_	AI_H_	NA_H_
Non treated RBCs	0.086 ± 0.04	131.5 ± 31.4	0.004 ± 0.001	11.3 ± 1.1
RBCs treated with 200 mM H_2_O_2_	0.073 ± 0.01 *	128.0 ± 13.3	0.005 ± 0.001	9.3 ± 2.8 *
RBCs treated with 300 mM H_2_O_2_	0.140 ± 0.03 *	114.0 ± 23.9	0.003 ± 0.002	8.4 ± 1.3 *
RBCs treated with 400 mM H_2_O_2_	0.173 ± 0.03 *	84.2 ± 25.3 *	0.005 ± 0.002	11.3 ± 2.5

* Indicates statistically significant difference (*p* < 0.05) in the values of H_2_O_2_-treated RBCs compared with those of non-treated RBCs.

## Data Availability

All the data is contained within the article.

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
