# Peer review of "Assessment of Red Blood Cell Aggregation in Preeclampsia by Microfluidic Image Flow Analysis—Impact of Oxidative Stress on Disease Severity"

_ijms, 2024, doi:10.3390/ijms25073732_

Round 1

Reviewer 1 Report

Comments and Suggestions for Authors

The study of Alexandrova-Watanabe and co-authors is devoted to the investigation of red blood cell (RBC) aggregation in different rheological conditions, in the presence of reactive oxygen species for healthy of preeclampsia pregnant women. The authors have observed appearance of RBC aggregates in a flow chambers at high or low shear stress and have presented data that RBC aggregates dissolve at high shear. For patients with PE the authors demonstrate significantly higher RBC aggregation compared to healthy pregnant women. Altogether, the study is relevant for the purpose of PE early diagnosis. However, there is a number of questions and control experiments those should be addressed before the paper could be accepted for publication.

Major points

1.       In the RBC washing procedure all blood cells including platelets and leukocytes will appear in cell suspension, and as in some healthy donors platelets aggregate in response to EDTA (EDTA-dependent pseudothrombocytopenia), the effects of platelet aggregation should be taken into account.

2.       Were there vehicle control samples for Preparation of Oxidized RBCs? Also, it is unclear, why Fig. 3A is noticeably different from Fig. 1A– according to the legends they should represent the same conditions.

3.       It is well known, that the wall shear rate is the shear stress parameter that affects cell behavior. Therefore the values of shear stress should be given in {1/s} not in {dyn/cm2}, because recalculation requires the parameters of the solution viscosity. 5 dyn/cm2 seems to be too high for capillary blood shear stress, please provide appropriate references to the literature.

4.       It seems that the threshold and other numeric parameters for the assessment of RBC aggregation were adjusted for each experiment individually, which is not correct for the analysis of microscopy data.

5.       The RBC aggregates are highly unstable and could disaggregate within minutes. Therefore, the timing of image collection should be given. For example:” 5 min low shear – stop flow - 30 sec pause – 100 ms imaging – 5 min high shear – stop flow – 30 sec pause – 100 ms imaging”.

6.       A supporting table containing all given in Table 1 anonymous data for individual patients should be given. In addition, the gestational age (GA) for each analysis should be given as well as the GA for the diagnosis of PE. As of now it is not clear whether RBC aggregation was assessed before delivery or after.

7.       It is not obvious what structures the authors call “networks” and “extended branches”. Please provide additional images in enough quantity to ensure that the observed phenomenon is not an artifact or “cherry-picking”.

8.       To support the conclusion that RBC aggregation correlates with the decease severity some kind of correlation or clustering analysis should be performed.

Minor points

1.       Both RBC and erythrocyte are used to denote RBC, which is not correct for human studies

2.       Scale bars should be given in all microscopy images

3.       The number of investigated patients and the information whether the image is representative for the patient group should be given in Figure legends.

4.       For Figure 1 the meaning of white lines should be given in the legend.

5.       The numbers of patients in each group should be given in the legend of Table 2

6.       Typo: Line 490: М. icrofluidic

Author Response

Dear Reviewer,

We sincerely appreciate your very insightful and constructive comments and recommendations. We introduced essential changes in the manuscript according to your recommendations. Hereby, is a point-by-point reply to your comments. Changes made to the manuscript have been marked with track changes.

  1. In the RBC washing procedure all blood cells including platelets and leukocytes will appear in cell suspension, and as in some healthy donors platelets aggregate in response to EDTA (EDTA-dependent pseudothrombocytopenia), the effects of platelet aggregation should be taken into account.

Answer: In our study, we chose EDTA for the blood sampling according to [Baskurt et al,; International Expert Panel for Standardization of Hemorheological Methods. New guidelines for hemorheological laboratory techniques. Clin Hemorheol Microcirc. 2009;42(2):75-97.] as the most widely used anticoagulant in hemorheological studies. We agree that EDTA-dependent pseudothrombocytopenia may affect RBC aggregation. However, upon current visual inspection, we don’t see platelets in our capillary after 3x PBS washes (see Methods section), moreover, according to the literature cited below, its incidence has been reported to be 0.09-0.21% in the literature, which is a very low percentage [Sakurai S.et al. Aminoglycosides prevent and dissociate the aggregation of platelets in patients with EDTA-dependent pseudothrombocytopenia. Br J Haematol. 1997; 99(4):817-823.; Zandecki M, et al. Spurious counts and spurious results on haematology analysers: a review. Part I: platelets. Int J Lab Hematol. 2007;29(1):4-20. Harmening D (2005) Modern blood banking and transfusion practices. Davis Company: Philadelphia.; Pitkin F (2017) Ethylenediaminetetraacetic Acid (EDTA)-Induced Thrombocytopenia: A Case Report. J Tradit Med Clin Natur 6: 209.; Akbayram S, et al. EDTA-dependent pseudothrombocytopenia in a child. Clin Appl Thromb Hemost. 2011 Oct;17(5):494-6. ].

Washing the cell suspension three times with PBS led to the depletion of platelet fraction, which was inspected by flow camera and on the cell smears with 50X magnification. PBS used for the washing does not contain EDTA (which has been cleared in the revised version of the manuscript).

The effect of platelet aggregation is undoubtedly an important aspect of the blood cell investigation; we have already initiated the experiments with platelets in a flow chamber.

  1. Were there vehicle control samples for Preparation of Oxidized RBCs? Also, it is unclear, why Fig. 3A is noticeably different from Fig. 1A– according to the legends they should represent the same conditions.

Answer: We thank the reviewer for these remarks. For the preparation of oxidized RBCs three other normotensive pregnant women were recruited who were not included in the control group PC. Now, it has been clarified in the revised version of the manuscript (sections 2.3. (line 428 in the pdf version); 4.1. (lines 781 – 783 - pdf); and 4.3. (line 824 - pdf)).

The images presented in Fig. 1A and Fig. 3A, are selected from the PC group and the group of the newly recruited normotensive pregnant women, respectively. Although the conditions under which the experiments were carried out were the same, there is some difference in the rheological characteristics between the individual samples in each group, reflected in the deviation from the mean value of the respective group (visible in Table 2 and Table 3). In the manuscript, selected images from each group were presented, corresponding to samples from different pregnant women, considering the minor differences between them.

  1. It is well known, that the wall shear rate is the shear stress parameter that affects cell behavior. Therefore the values of shear stress should be given in {1/s} not in {dyn/cm2}, because recalculation requires the parameters of the solution viscosity. 5 dyn/cm2 seems to be too high for capillary blood shear stress, please provide appropriate references to the literature.

Answer: According to the literature sources [Papaioannou TG, Stefanadis C. Vascular wall shear stress: basic principles and methods. Hellenic J Cardiol. 2005 Jan-Feb;46(1):9-15. PMID: 15807389], the shear stress in the capillary blood is lower or equal to 43 dyn/cm2 (and shear rate below 1250 s-1). The solution viscosity we measured at 37 ℃ is 1.12 (mPa.s). The highest shear stress, we applied in our experimental model in this study, in the microchannels of the BioFlux system, is 5 dyn/cm2 (recalculated shear rate – 446 s-1). To better clarify this point (according to the reviewer's suggestion), additional text was added in the “Introduction” (lines 179 – 182, pdf version): “Recent advances in microfluidic systems have led to more realistic models of blood vessels, used in many studies for blood flow analysis and appropriate selection of experimental flow conditions. These models applied the shear force effects on cell dynamics and their use in microfluidic applications” and the respective reference to the literature [40,41]”.

  1. It seems that the threshold and other numeric parameters for the assessment of RBC aggregation were adjusted for each experiment individually, which is not correct for the analysis of microscopy data.

Answer: In fact, the threshold process, which is a segmentation instrument from the Image J Software is applied automatically and there is no manual adjustment. The term “adjustment” was used in our previous version, as a software terminology, which is used as an automatic setting of the corresponding software tool. In the revised version this terminology is avoided to escape misunderstanding and was replaced with “software setting”.

  1. The RBC aggregates are highly unstable and could disaggregate within minutes. Therefore, the timing of image collection should be given. For example:” 5 min low shear – stop flow - 30 sec pause – 100 ms imaging – 5 min high shear – stop flow – 30 sec pause – 100 ms imaging”.

Answer: We thank the reviewer for the recommendation. According to this remark, additional text was added in section 4.7. “Design of the experiments” (lines 886 – 888, pdf): “After low and high shear rates (each of them with a duration of 5 min), 30 s after stopping the flow, at least 5 images were taken every 2 s along the entire visible length of the channel at randomly selected locations”.

  1. A supporting table containing all given in Table 1 anonymous data for individual patients should be given. In addition, the gestational age (GA) for each analysis should be given as well as the GA for the diagnosis of PE. As of now, it is not clear whether RBC aggregation was assessed before delivery or after.

Answer: According to your suggestion, additional data containing all indices given in Table 1 for individual patients are provided (Table S1 and Table S2, supplementary material).

All the experiments were carried out with blood taken at the time of diagnosis. Therefore, in section 4.2. “Sample Preparations”, it has been noted (lines 812 – 814): ”Blood from patients with preeclampsia was collected immediately after diagnosis before treatment was prescribed and by normotensive pregnant women at their prenatal visits.”. Now to better clarify this point, i.e., to precise the time of RBC isolation for all experiments, according to your suggestion in the revised version additional text was added in section 2.2 (lines 289 – 290, pdf version): “Red blood cells from the blood of patients with PE were isolated at the time of diagnosis before patients received any treatment”.

  1. It is not obvious what structures the authors call “networks” and “extended branches”. Please provide additional images in enough quantity to ensure that the observed phenomenon is not an artifact or “cherry-picking”.

Answer: To better clarify the type of the aggregate formations, i.e. branched aggregates and extended branches (network), a new figure (Fig. 2) was added to the text. To additionally demonstrate that the observed phenomenon is not an artifact several new images of RBC aggregates, representative of controls and patients with PE were presented in Supplementary material (Fig. S1 – S3).

  1. To support the conclusion that RBC aggregation correlates with the decease severity some kind of correlation or clustering analysis should be performed.

Answer: Determining the disease severity is based on a complex assessment of the patient’s status, which includes, on the one hand, the presence of increased blood pressure and/or proteinuria, but also dysfunction in some organs, vertigo, headaches, edema, etc. To support the conclusion that RBC aggregation correlates with disease severity, in the revised version we applied Pierson’s correlation analysis between blood pressure and aggregation indices of the studied groups of patients. In this regard, additional text was added in the “Discussion“ section (lines 570 – 573; and 577 – 597), and a table (Table S3) in Supplementary material, respectively.

Minor points

  1. Both RBC and erythrocyte are used to denote RBC, which is not correct for human studies

Answer: It is considered and corrected in the revised version.

  1. Scale bars should be given in all microscopy images

Answer: Scale bars have been added to all microscopy images in the revised version.

  1. The number of investigated patients and the information whether the image is representative of the patient group should be given in Figure legends.

Answer: In the revised version this information has been added.

  1. For Figure 1 the meaning of white lines should be given in the legend.

Answer: These white lines act as separators to distinguish between the different microscope images (i.e. between the image of Fig. 1A and Fig. 1B; and between that in Fig. 1B and Fig. 1C). Now it is mentioned in the legend.

  1. The number of patients in each group should be given in the legend of Table 2

Answer: In the revised version, the number of patients in each group has been added.

  1. Typo: Line 490: М. Icrofluidic

Answer: Now it is corrected.

Reviewer 2 Report

Comments and Suggestions for Authors

In this manuscript the author used microfluidics techniques and new image flow analysis to evaluate RBC aggregation in preeclamptic and normotensive pregnant women.

The research topic of this manuscript is meaningful, but it does not meet the high standards of the journal (INTERNATIONAL JOURNAL OF MOLECULAR SCIENCES). In this manuscript, a lot of space is devoted to introducing the conclusions of previous studies. The author only analyzed the red blood cell aggregation of three kinds of pregnant women (normal, non-severe eclampsia, severe eclampsia) through microscopic images, and the innovative contribution made by the author is limited. In addition, there are some issues that were not clearly explained in the manuscript. So, my opinion is to suggest the author to submit to another journal instead.

1. How to define whether red blood cells are in an aggregated or non-aggregated state(how many cells stick together is defined as aggregated, and how to calculate)?

2. How are the values of ‘S2’ and ‘S3’ obtained in EAL calculation, and how to define whether a pixel belongs to normal red blood cells or aggregated red blood cells?

3. How to calculate NEA?

4. What conclusions can the authors draw from treating red blood cells with hydrogen peroxide? (Table 3)

5. Table 3 shows why, compared with untreated red blood cells, the number of red blood cell aggregation decreases under low shear force and increases under high shear force after treatment with 200Mm hydrogen peroxide.

6. The title of the manuscript mentions the use of microfluidic analysis, but the article does not elaborate on the structure of microfluidic chips or provide system photos.

7. Is microfluidic chips absolutely necessary in this study?

Comments on the Quality of English Language

well.

Author Response

Dear Reviewer,

We deeply appreciate your very constructive recommendations. Hereby a point-by-point reply to your comments. Changes made to the manuscript have been marked with track changes.

  1. How to define whether red blood cells are in an aggregated or non-aggregated state (how many cells stick together is defined as aggregated, and how to calculate)?

Answer: The differentiations of the state of the cells (aggregate or non-aggregate) were determined according to the number of the stuck together /(clumped) cells one over the other and we have chosen as an initial parameter (minimum) 3 grouped cells. To clarify this differentiation additional text was added in 4.8. (lines 939 – 942, pdf version of the revised manuscript). The automatically incorporated functions of the software environment ImageJ realize the calculation. Based on this parameter, the initial assessment of RBC aggregates is performed. It verifies that all the RBC aggregates are selected.

A detailed explanation of the processes is given in the answer to question Number 3.

  1. How are the values of ‘S2’ and ‘S3’ obtained in EAL calculation, and how to define whether a pixel belongs to normal red blood cells or aggregated red blood cells?

Answer: The value of S2 (in pixels) is the total sum of areas of the aggregates, obtained in the low shear rate from one visual field (as can be seen in Fig.6D). The value of S3 (in pixels) is the total sum of areas of the aggregates, obtained in the high shear rate from one visual field. The visual field is fixed for all the experiments with the area (S1) of 1600 x 300 pixel2 and additional text was added in section 4.9. line 960. 

The differentiation in pixels between the normal red blood cells (non-aggregated) and aggregated red blood cells is realized by choosing the criteria parameter. This parameter (in pixels) corresponds to the area of 3 stuck together /(clumped) cells.

  1. How to calculate NEA?

Answer: To calculate the number of erythrocyte aggregates (NA), several steps (which are defined in this study from section “4.8. Computational image analysis for the evaluation of the RBC aggregates” and are illustrated in Fig. 6), are realized as follows:

-The obtained raw images of erythrocyte aggregates are imported as input data into the Image J software environment.

- The brightness/contrast tool of the software is applied to improve the contrast of the images.

-The next step includes the application of thresholding for segmentation: it involves software automatic setting of the image's brightness levels to segregate the RBC aggregates from the rest of the image.

-The next step is the selection of the criteria parameter (pixels corresponding to aggregates including 3 cells, combined/stuck one over the other), which determines the separation between erythrocyte aggregates and non-aggregates. Based on this parameter, the initial assessment of RBC aggregates is performed. It verifies that all the RBC aggregates are selected.

- If this parameter is fulfilled/met, the next step is to obtain the table with results for the size of the area of each object of interest and their number (i.e., aggregates).

- The obtained results for the number and the area size of the RBC aggregations (total numbers and size of RBC aggregates) were calculated in a program elaborated in IntelliJ IDEA.

  1. What conclusions can the authors draw from treating red blood cells with hydrogen peroxide? (Table 3)

Answer:  It is notable that under hypoxic conditions, partial oxygenation of Hb takes place, increasing autoxidation, which leads to ROS production near the RBC membrane. Aiming to better understand the impact of hypoxia and oxidative stress on PE pathology, we studied the impact of in vitro chemically induced oxidative stress on RBC.

According to your remark, an additional explanation was added in section 2.3. (lines 536 – 551): “At the lowest concentration of H2O2 tested (200 mM), the aggregation index of red blood cells was lower, while the number of aggregates was not significantly different from that of untreated cells. The increasing H2O2 concentration at a low shear rate leads to an elevation in both AI and the number of RBC aggregates compared to those of the untreated ones (Table 3). The Pearson's correlation analysis we applied demonstrated a strong relation (r=0.9; p=0.033) for AIL of the pair parameters of treated cells with 300 mM H2O2 and those derived from the PE non-severe group and moderate (r=0.55; p=0.05) for RBCs treated with 400 mM H2O2 and RBCs from the PE severe group. Although the oxidative stress model we applied in this study does not fully account for disease progression it sheds light on the influence of oxidative stress on the RBC rheological behavior.

At a high shear rate, the aggregation index of RBCs treated with the three H2O2 concentrations did not differ significantly from that of untreated cells. The number of RBC aggregates remaining intact at high shear flow conditions treated with the lower concentrations of 200 and 300 mM H2O2 was lower compared to that at the highest concentration of H2O2 (400 mM), and to that of non-treated RBCs, which we hypothesize is a reflection of intact antioxidant defense machinery in healthy individuals.”

  1. Table 3 shows why, compared with untreated red blood cells, the number of red blood cell aggregation decreases under low shear force and increases under high shear force after treatment with 200Mm hydrogen peroxide.

Answer: At low shear flow conditions (after treatment with 200Mm), the average number of RBC aggregates decreases slightly compared with untreated ones, but without a statistically significant difference. At high shear flow conditions, the average number of RBC aggregates was lower than in the untreated cells (Table 3). The plausible explanation for this phenomenon we expanded in the revised version of the manuscript in the “Discussion“, section 3.2. (lines 696 – 702): “An interesting finding, we made was that at low concentration of H2O2 (200 mM), the degree of aggregation and aggregation indices were somewhat less expressed compared to untreated cells. In line with this finding, it was previously demonstrated a slightly increased RBC membrane deformability in response to H2O2 treatment [ Sinha A et al. Single-cell evaluation of red blood cell bio-mechanical and nano-structural alterations upon chemically induced oxidative stress. Sci Rep. 2015, 7;5:9768.]. As noted above, RBC deformability largely determines the formation of aggregates, altering rheological blood properties. Another plausible hypothesis is that low-dose oxidant incubation triggers the innate defense RBC mechanisms that are intact in healthy cells and elicit a potent response.”

  1. The title of the manuscript mentions the use of microfluidic analysis, but the article does not elaborate on the structure of microfluidic chips or provide system photos.

Answer: According to your remark, the title of the manuscript has been corrected to reflect the new image-processing methodology developed.

  1. Is microfluidic chips absolutely necessaryin this study?

Answer: To effectively track the movement of the aggregates, a plate (chip) with microchannels with dimensions of 350 × 75 μm is used, in which aggregates are limited to move in only one direction. Another reason for using this kind of microfluidic chip is the possibility of connecting to an electro-pleural pump, with the help of which the shear rates in the channels can be varied very precisely. 

Reviewer 3 Report

Comments and Suggestions for Authors

1. Please add a scale bar to all images. Even if exact measurements are unavailable, an approximate scale based on other data will greatly aid in understanding the physics/engineering aspects of the problem.

2. A citation is required at line 48 to support the statement concerning an unresolved problem in the field.

3. At line 216, please correct the typo from hear to shear.

4. Does the algorithm account for out-of-plane aggregates? If so, what is the probability of aggregates forming one above or below another? Additionally, how does the algorithm manage blurred regions within the images?

5. What is the viscosity of the medium/buffer used in the experiments? Is it possible to specify this numerically?

6. The manuscript frequently uses the term simulation. Consider whether emulate might be a more accurate term, particularly when referring to hardware mimicry, as opposed to simulation, which is often used for software processes. A thoughtful consideration of terminology is suggested.

7. Please provide explicit definitions of high shear flow and low shear flow, including references for these terms.

8. Comment on how the viscoelasticity of the aggregates influences the results, including any variations with respect to flow rate.

9. In real-world conditions, channel walls may deform, leading to complex effects on flow. Has this factor been considered in the study, and if so, would the results remain consistent? Have any studies addressed this phenomenon?

Comments on the Quality of English Language

Please correct the typos and, if possible, use short sentences that are easy to read for people outside the field. Use abbreviations wherever necessary to avoid repetitions of the words.   

Author Response

Dear Reviewer,

We deeply appreciate your very insightful and constructive comments and recommendations. These are our point-by-point responses to your comments. Changes made to the manuscript have been marked with track changes.

  1. Please add a scale bar to all images. Even if exact measurements are unavailable, an approximate scale based on other data will greatly aid in understanding the physics/engineering aspects of the problem.

Answer: We thank the reviewer for this remark. In the revised version, scale bars have been added to all microscopy images.

  1. A citation is required at line 48 to support the statement concerning an unresolved problem in the field.

Answer: A new citation [2] has been added according to your recommendation.

3. At line 216, please correct the typo from hear to shear.

Answer: It is corrected in the revised version.

4. Does the algorithm account for out-of-plane aggregates? If so, what is the probability of aggregates forming one above or below another? Additionally, how does the algorithm manage blurred regions within the images?

Answer:

The study's objectives and methodologies are focused on different aspects concerning flow image analysis using ImageJ, which primarily involves the 2D analysis and quantification of aggregates within the plane of the image.

The second part of the question is connected with the functions of the software environment Image J. There are specific software tools that allow to control the blurred regions- like "DeconvolutionLab2" and "Unsharp Mask" plugins are used for managing heavily blurred regions. In this specific study, the obtained images are of such quality that they do not require using the above-mentioned plugins, but only the integrated into this software “Brightness/Contrast” instrument.

  1. What is the viscosity of the medium/buffer used in the experiments? Is it possible to specify this numerically?

Answer: We found that the viscosity of the medium used in the experiments was 1.12 mPa.s. A new section 4.5. “Viscosity measurements” (lines 836 – 841, pdf version of the revised manuscript) was added in 4. Materials and Methods.

  1. The manuscript frequently uses the term simulation. Consider whether emulate might be a more accurate term, particularly when referring to hardware mimicry, as opposed to simulation, which is often used for software processes. A thoughtful consideration of terminology is suggested.

Answer: According to your suggestion in the revised version of the manuscript we have changed the term” simulation of oxidative stress” witch “chemically induced oxidative stress”.

7. Please provide explicit definitions of high shear flow and low shear flow, including references for these terms.

Answer:  According to your remark, in the revised version we provide explicit definitions of high shear flow and low shear flow (“Introduction” section, lines 141; 145 – 146, pdf version): “In normal conditions, in large vessels where blood velocity is low, with a lower shear rate (<1-10 s-1), red blood cells tend to aggregate [22]” and “At high shear rates (>150 s-1) in large vessels RBCs’ deformability and dissociation results in low whole blood viscosity [23].”

  1. Comment on how the viscoelasticity of the aggregates influences the results, including any variations with respect to flow rate.

Answer: According to your suggestion, additional text was added in the “Discussion” section (lines 669 – 674, pdf version): “Blood viscoelastic properties are particularly sensitive indicators of aggregation and stiffness of RBCs. At a low shear rate (<10 s-1) the viscoelastic properties depend mainly on the RBC aggregation [Hahn R et al. Viscoelasticity and Red Blood Cell Aggregation in Patients with Coronary Heart Disease. Angiology. 1989;40(10):914-920.]. Our results demonstrated enhanced RBC aggregation in PE cases, especially in the severe group relative to the controls. Therefore, the elevated RBC aggregation in PE groups could be caused by the impaired viscoelasticity of RBCs in PE patients.

  1. In real-world conditions, channel walls may deform, leading to complex effects on flow. Has this factor been considered in the study, and if so, would the results remain consistent? Have any studies addressed this phenomenon?

Answer: In most studies, simulations/models of blood flow assume rigid walls, which is due to the difficulty in solving the problem of the relationship between blood flow and vessel wall deformation. In this regard, computational models could contribute to a better understanding of the relation between the model of blood flow and vessel wall dynamics in real-world conditions, especially in patients with preeclampsia and patients with high blood pressure. In this regard, significant progress has been made in solving the problem of blood flow in deformable walls using the Arbitrarily Lagrangian-Eularian method [Gerbeau JF, et al. Fluid–structure interaction in blood flows on geometries based on medical imaging. Computers and Structures. 2005; 83(2–3):155–165.; Bazilevs Y, et al. Isogeometric fluid–structure interaction analysis with applications to arterial blood flow. Computational Mechanics. 2006; 38(4–5):310–322.], the immersed boundary method [Kim Y. et al. Blood flow in a compliant vessel by the immersed boundary method. Annals of Biomedical Engineering. 2009; 37(5):927–942.], and Coupled Momentum method (CMM) for fluid–solid interaction to model blood flow and vessel wall dynamics [Xiong G et al. Simulation of blood flow in deformable vessels using subject-specific geometry and spatially varying wall properties. Int J Numer Method Biomed Eng. 2011 Jul;27(7):1000-1016.].

The application of the computational model to the effect of the deformable  channel walls on the blood flow would be an interesting topic for our future research in this direction but was not the subject of the present work.

Comments on the Quality of English Language

Please correct the typos and, if possible, use short sentences that are easy to read for people outside the field. Use abbreviations wherever necessary to avoid repetitions of the words. 

Answer: Now it is corrected.  

Round 2

Reviewer 1 Report

Comments and Suggestions for Authors

In the revised version, the authors responded quite satisfactorily to all my comments and took into account all the recommendations.

Author Response

Comments and Suggestions for Authors

In the revised version, the authors responded quite satisfactorily to all my comments and took into account all the recommendations.

Answer:

Dear Reviewer,

We deeply appreciate your very constructive comments and recommendations. Thanks to your recommendations, the manuscript has been significantly improved.

Reviewer 2 Report

Comments and Suggestions for Authors

The author's revised version answered my question well and agreed to be published. There are some minor issues that need to be addressed.

1、Do not include a period in the title.

2、The supplementary materials are not detailed, and there should be necessary explanations for the images. In addition, some important methods should also be added to the supplementary materials, such as the calculate of NEA.

Comments on the Quality of English Language

well

Author Response

Reviewer 2

The author's revised version answered my question well and agreed to be published. There are some minor issues that need to be addressed.

We thank the Reviewer for her/his valuable comments. Changes made to the manuscript and the Supplementary Material have been marked with track changes.

1、Do not include a period in the title.

Answer: Now it is corrected.

2、The supplementary materials are not detailed, and there should be necessary explanations for the images. In addition, some important methods should also be added to the supplementary materials, such as the calculate of NEA.

Answer: The Supplementary Materials are systematized by sections, corrected in the revised version. Additional explanations are provided for each of the images presented (S1 – S3). A detailed description of the quantification of the number of RBC aggregates (NA) is given in the "Methods" section of the Supplementary material.